# Baculovirus Display of Varicella–Zoster Virus Glycoprotein E Induces Robust Humoral and Cellular Immune Responses in Mice

**DOI:** 10.3390/v14081785

**Published:** 2022-08-16

**Authors:** Wenhui Xue, Tingting Li, Sibo Zhang, Yingbin Wang, Minqing Hong, Lingyan Cui, Hong Wang, Yuyun Zhang, Tingting Chen, Rui Zhu, Zhenqin Chen, Lizhi Zhou, Rongwei Zhang, Tong Cheng, Qingbing Zheng, Jun Zhang, Ying Gu, Ningshao Xia, Shaowei Li

**Affiliations:** 1State Key Laboratory of Molecular Vaccinology and Molecular Diagnostics, School of Life Sciences, School of Public Health, Xiamen University, Xiamen 361102, China; 2National Institute of Diagnostics and Vaccine Development in Infectious Diseases, Xiamen University, Xiamen 361102, China

**Keywords:** Varicella Zoster Virus (VZV), HZ vaccine, glycoprotein E, baculovirus display, cellular immunity

## Abstract

Varicella–zoster virus (VZV) is the causative agent of varicella and herpes zoster (HZ) and can pose a significant challenge to human health globally. The initial VZV infection—more common in children—causes a self-limiting chicken pox. However, in later life, the latent VZV can become reactivated in these patients, causing HZ and postherpetic neuralgia (PHN), a serious and painful complication. VZV glycoprotein E (gE) has been developed into a licensed subunit vaccine against HZ (Shingrix). However, its efficacy relies on the concomitant delivery of a robust adjuvant (AS01B). Here, we sought to create a new immunogen for vaccine design by displaying the VZV–gE on the baculovirus surface (Bac–gE). Correct localization and display of gE on the engineered baculovirus was verified by flow cytometry and immune electron microscopy. We show that Bac–gE provides excellent antigenicity against VZV and induces not only stronger gE-specific CD4^+^ and CD8^+^ T cell responses but also higher levels of VZV–specific neutralizing antibodies as compared with other vaccine strategies in mice. Collectively, we show that the baculovirus display of VZV–gE confers ideal humoral and cellular immune responses required for HZ vaccine development, paving the way for a baculovirus-based vaccine design.

## 1. Introduction

Varicella–zoster virus (VZV; also known as human herpesvirus 3) causes chickenpox (varicella) during primary infection and establishes a lifelong latent infection in the ganglia of the peripheral nervous system [1]. Typically, varicella is a self-limiting disease, occurring most frequently in children. However, the latent virus shows a high risk of reactivation later in life as herpes zoster (HZ)\shingles [2], particularly among older people or immunocompromised patients. HZ causes a severe and painful rash, with postherpetic neuralgia (PHN) considered a serious complication in many patients [3].

HZ incident rates hover around 3–4 cases per 1000, with a comparable prevalence in males and females, but PNH is higher in females [4,5]. Worldwide, the disease burden of HZ increases with age, and is speculated to cause at least 2.77 million cases annually in China [6]. Some of the symptoms associated with HZ can be alleviated with antivirals that are typically used against herpes viruses, such as acyclovir, famciclovir and valaciclovir. However, these antivirals have a limited effect on HZ and none of them can prevent reactivation of latent VZV [7,8]. HZ onset is thought to be caused by a declining VZV–specific cell–mediated immunity (CMI). Indeed, in many older patients, there is a strong bidirectional link between HZ and age-related functional impairment, which manifests as an inability or decline in the capacity to perform tasks associated with daily living [9,10]. Consequently, treatments that can enhance a VZV–specific CMI response will be important considerations in the treatment and prevention of HZ [11]. Thus, effective HZ vaccines that address this decreased immunity are highly desired.

The first live-attenuated varicella vaccine—the Oka strain—was developed by Takahashi and colleagues in 1974 [12]. This vaccine—later developed and marketed as Varivax by Merck—has been used for the safe immunization of children worldwide. Since then, two vaccines against shingles have been developed, indicated for adults over 50 years: Zostavax (Merck), which also contains the live-attenuated varicella virus but in higher concentrations than in Varivax, and Shingrix (GlaxoSmith Kline (GSK)), which comprises a recombinant glycoprotein E combined with a powerful AS01B adjuvant that elicits a CMI response. Shingrix has an efficacy of 97.4% for the prevention of HZ in clinical trials in 60- to 69-year-olds, which is considerably higher than that achieved by Zostavax (64%) [13,14]. Despite the efficacy, Shingrix vaccinations are associated with extreme discomfort; the percentage of grade 3 adverse reactions reach up to 5–10% in adults aged 50 and above, as reported in a clinical data [15,16].This might be largely ascribed to the AS01B adjuvant, which contains two immune stimulates: QS21 and MPL. QS-21 could activate antigen-presenting cells (APCs) and T cells and stimulate Th2 humoral and Th1 cell–mediated immune responses and the liposomes (AS01) formed by monophosphate lipid A (MPL A); and they both have a synergistic mechanism in boosting vaccine immunogenicity [17]. QS21 is a triterpene saponin purified from *Quillaja saponaria*. Although QS-21 has been used for several vaccines in clinical studies, there are some innate drawbacks of this natural product such as chemical instability, scarcity, heterogeneity and dose-limiting toxicity which have limited its extensive clinical applications. Additionally, the high price of Shingrix vaccines means it is not realistically accessible for developing countries. Therefore, developing an alternative new HZ vaccine platform could help to prevent and control the HZ disease.

The baculovirus expression vector system is a well-established platform for the production of recombinant proteins and vaccines [18,19]. Among various Baculoviridae family members, *Autographa californica* multicapsid nucleopolyhedrovirus (AcMNPV) is the most extensively studied and has a complete annotated genome sequence [20,21]. Several vaccine candidates using baculovirus have since been developed and are in preclinical trials against zika virus, influenza virus, Porcine circoviruses and SARS Coronavirus [22,23,24,25]. Of particular note, baculoviruses trigger the innate immunity and evoke robust humoral and cellular immune responses due to the abundance of bioactive CpG motifs in the genome [26]. These CpG elements act as agonists of Toll-like receptor 9 (TLR9), conferring strong adjuvanticity and promoting robust activation of the innate immune system by the stimulation to dendritic cells [27]. Thus, the baculovirus itself may be an effective adjuvant. Unlike other commonly working vectors from mammalian viruses, baculoviruses show no preexisting immunity in the herd of humans without circulating because of host barrier; consequently, this can avoid neutralization against the baculovirus vector by potential preexisting antibodies [28]. For these reasons, the recombinant baculovirus is regarded as an excellent vector tool for vaccine development.

Here, we developed a novel HZ vaccine candidate based on the recombinant baculovirus display of gE (Bac–gE). We anticipated that the antigenic display on the surface of the recombinant baculovirus will lead to amplified B cell responses. In addition, because the construct contains the hybrid human promoter for antigenic expression in mammalian cells, we expect that the recombinant baculovirus will also induce antigen-specific T cell immune responses [29]. Through our analyses, we show that immunization with Bac–gE in mice induces robust gE-specific neutralizing antibodies as well as gE-specific CD4^+^ and CD8^+^ T cell responses in the absence of an additional adjuvant. This strategy provides an alternative vaccine candidate to address the immunosenescence issue common among patients with HZ, particularly those with a compromised immunity.

## 2. Materials and Methods

### 2.1. Cell, Virus Stock, Reagents and Antibodies

*Spodoptera frugiperda* (Sf9) insect cells (Thermo Fisher Scientific; Waltham, MA) were maintained in suspension in serum-free ESF921 medium (Thermo Fisher Scientific) at 28 °C. Human retinal epithelial ARPE-19 cells (The American Type Culture Collection [ATCC]; Manassas, VA) were plated into 24-well plates and grown in DMEM/F12 (1:1) medium (Thermo Fisher Scientific) containing 10% fetal bovine serum (FBS; Hyclone, Logan, UT). HepG2 (ATCC) cells were cultured in DMEM with 10% FBS and used for baculovirus infection. The live-attenuated viral vaccine (v-Oka strain from ATCC; AR-795, prepared in BSL 2+ facility of Beijing Wantai Biopharm according to the modality of a varicella vaccine) was stored at −80 °C and was titrated using a plaque assay before experimentation. The recombinant baculovirus was constructed using the BacMagic system (Merck & Co Inc, Kenilworth, NJ, USA) and the progeny virus was stored at 4 °C. gE-specific monoclonal antibody, 1B11 [30] and HRP-conjugated secondary antibodies were purchased from Thermo Fisher Scientific. The BacPAK Baculovirus Rapid Titer Kit for determining the titer of the baculovirus stock was purchased from Takara (Beijing, China).

### 2.2. Construction of Recombinant Baculovirus

The nucleotide sequence encoding the ectodomain of glycoprotein E (NC_001348) was synthesized by Shangon BioTech (Shanghai, China). To enhance the display of gE on the baculovirus, the construct contained the gp64 signal peptide and a polyhistidine (6 × His) tag at the N-terminus, a transmembrane region, and the C-terminal domain of gp64 at the C-terminus. The construct was cloned into the pTriex-5 plasmid (Merck), which harbors a dual promoter (CMV for expression in mammalian cells and P10 promoter for expression in insect cells). The pTriex-gE plasmids were then co-transfected with circular baculovirus DNA into Sf9 cells to produce the active recombinant baculovirus. After four rounds of subculture, supernatants were collected for subsequent experiments. The empty pTriex vector was used to produce the wild-type baculovirus using the same procedures.

### 2.3. gE Plasmids Construction and Protein Purification

The glycoprotein E ectodomain sequence (NC_001348) was synthesized and cloned into a baculovirus vector pTriex-5 by Shangon BioTech (Shanghai, China). gE protein expression and purification were performed as described previously [31]. In brief, gE plasmids were co-transfected with linearized 2.0 DNA (deficient in v-cath/chiA genes) (Expression Systems, CA, USA) into Sf9 insect cells (Thermo Fisher Scientific, Waltham, MA, USA), according to the protocol provided by the manufacturer (Expression Systems). After four rounds of subculture, supernatants were collected. For purification, the supernatant was centrifugated at 7000 rpm for 15 min and dialyzed against phosphate-buffered saline (PBS), pH 7.4, then purified with Ni-sepharose fast-flow 6 resin (GE Healthcare, Boston, MA, USA), and eluted with 250 mM imidazole (Appendix A). The protein concentrations of the final purified samples were measured with Pierce BCA Protein Assay Kit (Thermo Fisher Scientific). The reactivities of purified proteins were tested using an indirect enzyme-linked immunosorbent assay (ELISA) (Appendix A).

### 2.4. Purification and Titration of the Recombinant Baculovirus

After three days of infection, supernatants containing the baculovirus were collected and roughly purified to remove cell debris by centrifugation at 7000× *g* using a JA-10 rotor (Beckman Coulter). To increase the purity of the recombinant baculovirus, the supernatant was further centrifuged at 10,000× *g* using a JA-10 rotor (Beckman Coulter). The supernatant was discarded, and the tube containing the pelleted material was inverted onto a paper towel to remove residual liquid. The pellets were then resuspended with Tris-HCl (PH 7.0) and stored at 4 °C. Baculovirus titers were determined using the BacPAK Baculovirus Rapid Titer Kit, according to manufacturer’s instructions.

### 2.5. Western Blotting

Equal amounts of Sf9 cell lysates were loaded onto denaturing SDS-PAGE gels and electrophoresed for 90 min at 80 V in a BioRad MINI-PROTEAN Terea system (BioRad Laboratories; Irvine, CA, USA). GAPDH served as a control for equal protein loading. Gels were then transferred onto nitrocellulose membranes (Whatman; Dassel, Germany) using a Trans-Blot Turbo transfer system (Bio-Rad Laboratories). Membranes were blocked at room temperature for 1 h and then incubated with the 1B11 gE-specific antibody. After three washes with PBST (0.1% Triton-100 in PBS), membranes were incubated with horseradish peroxidase (HRP)-conjugated goat anti-mouse secondary antibodies at room temperature for 30 min. Membranes were washed again and the proteins were detected using the SuperSignal West Pico Chemiluminescent Substrate (Thermo Fisher Scientific).

### 2.6. Immunofluorescence Assay (IFA)

Immunofluorescence staining was visualized using a fluorescence microscope (EVOS M7000; Thermo Fisher Scientific), as described previously [32]. Sf9 cells were seeded into the wells of 24-well plates and infected with recombinant baculovirus at an MOI of 1. At 48 h after infection, cells were fixed with 4% paraformaldehyde for 20 min and permeabilized with 0.2% Triton X-100 for 20 min. After several washes with PBS, the samples were blocked with 5% bovine serum albumin at 37 °C for 30 min and incubated with gE-specific antibody (1B11, 1:2000) overnight at 4 °C. After three washes, the samples were incubated with Alexa Fluor 647-conjugated goat anti-mouse IgG secondary antibody (1:1000 dilution) at 37 °C for 1 h. Nuclei were counterstained with DAPI. Fluorescence was visualized using a fluorescence microscope and images were analyzed with Fiji (Image-J) software (NIH; Bethesda, MD, USA). Bac–WT cells were treated in the same manner.

### 2.7. Flow Cytometry

Sf9 cells (~2 × 10^6^) were infected with Bac–gE or Bac–WT recombinant baculovirus and cultured at 27 °C for 3 days. Cells were harvested by centrifugation at 500× *g*, seeded into 96-well U-bottom plates, and stained with gE-specific 1B11 antibody. After washing with PBST, cells were then counterstained with secondary antibody coupled to Alexa Fluor 647 for 1 h. Single-cell suspensions were then prepared for flow cytometry, with the data analyzed using FlowJo software (BD Biosciences; Franklin Lakes, NJ, USA). Cells were gated by forward scatter (FSC) and side scatter (SSC) to obtain single cells, and then gated for Alexa Fluor 647 positive versus negative staining. Cells without fluorescence staining were used as negative controls.

### 2.8. Enzyme-Linked Immunosorbent Assay (ELISA) 

gE proteins purified from Sf9 cells were coated onto the wells of 96-well plates at 100 ng per well and incubated overnight at 4 °C. Wells were then blocked with 2% BSA at 37 °C for 2 h. Sera were three-fold serially diluted from a starting concentration of 1:100 (100 µL) and added to the wells for 1 h at 37 °C. The plates were then washed and incubated with HRP-labeled secondary antibody (1:5000) for 1 h. Wells were then washed thrice, and the cells incubated with 100 µL of peroxidase substrate (o-phenylenediamine dihydrochloride; Sigma-Aldrich). The reaction was quenched after 10 min at 37 °C by the addition of 2 M sulfuric acid. Reactions were read on a microplate spectrophotometer (Tecan; Männedorf, Switzerland) at OD450/620 nm. For sandwich ELISA, the gE-specific antibody 4A2 was coated onto the wells of ELISA plates (100 ng/well) and incubated overnight at 4 °C. The plates were then blocked at 37 °C for 2 h and then incubated with the gE protein. Following a 1 h incubation, the plates were washed 5 times with PBST buffer and then incubated for 1 h with HRP-labeled 6H7 (Appendix A). The plates were again washed 5 times with PBST buffer and incubated with TMB substrate solution for 10 min. Finally, the reaction was stopped with 2 M H_2_SO_4_ and read at OD450/620 nm on a microplate reader. The data were analyzed using GraphPad Prism software 8.0 (GraphPad; San Diego, CA, USA).

### 2.9. Immune Electron Microscopy (IEM)

Recombinant baculovirus particles were fixed onto carbon-coated grids with immunogold label and visualized by negative-staining electron microscopy, as previously described [33]. Briefly, glow-discharged and carbon-coated grids were floated onto 10 µL drops of sample for 30 min at room temperature. The grids were placed on 20 μL of 1% glutaraldehyde droplets for 5 min and washed 5 times with 100 μL of ultrapure water. After washing, the grids were blocked and then incubated with gE-specific 1B11 antibody (1:50) overnight at 4 °C. Grids were then incubated with rabbit anti-mouse IgG conjugated with 5 nm gold particles (1:20 dilution) for 1 h. The grids were washed with distilled water, negatively stained with 2% phosphotungstic acid, and then examined with a transmission electron microscope (FEI; Thermo Fisher Scientific).

### 2.10. Immunization of BALB/c Mice

Six-week-old syngeneic BALB/c female mice (specific pathogen-free, SPF) were divided randomly into 6 groups (5 mice per group) and immunized with one of the following: Bac–gE combined with or without aluminum adjuvant, gE with aluminum, v-Oka vaccine, Bac–WT without adjuvant, or PBS. All vaccine modalities contained 0.3 μg gE proteins, as confirmed by sandwich ELISA. The detailed immunization scheme is shown in Table 1. Mice were immunized via an intramuscular (i.m.) route with 100 μL vaccine administered at 0 and 21 days. Blood samples were harvested weekly and centrifuged at 13,000× *g* for 10 min. Serum samples were preserved at −20 °C before analysis. Splenocytes were collected on day 5 post-immunization for enzyme-linked immunospot assay (ELISPOT) and intracellular cytokine staining (ICS) measurements.

### 2.11. Neutralization Assay

VZV–specific neutralizing antibody titers were determined using a 50% plaque reduction neutralization assay (PRNT50), as described previously, with some modifications [34]. Briefly, serum samples were heat-inactivated at 56 °C for 30 min before use. Next, 2-fold serially diluted mouse sera (initial dilution of 1:100) were mixed with an equal volume of 30 µL v-Oka strain virus (Wantai, Beijing, China) and incubated at 37 °C for 1 h. Subsequently, the cultivated mixture was added to the ARPE-19 cells and incubated for 1 h. Then, the supernatant was replaced with 1 mL of fresh DMEM\F12 medium, and the cells were maintained in a 37 °C incubator for 72 h. Enzyme-linked immunospot (ELISPOT) assay was performed according to standard laboratory protocols, with minor modifications. Spots were scanned with an ELISpot Reader (Cellular Technology Limited; Shaker Heights, OH) and calculated with the ImmunoSpot Analyzer (Cellular Technology Limited). Neutralization titers were determined as the highest serum dilutions that could neutralize half of the virus.

### 2.12. Enzyme-Linked Immunospot Assay (ELISPOT)

Enzyme-linked immunospot assay (ELISPOT) was performed according to the manufacturer’s instructions (MabTech, Sweden). Briefly, 15-mer peptides of VZV–gE were prepared: each peptide had an 11-amino acid overlap to cover the entire sequence of the gE protein. For the assay, an optimal number (5 × 10^5^ cells per well) of single cells were inoculated onto precoated ELISPOT plates (MabTech). Cells were then incubated with pooled VZV–gE peptides and stimulated for 20 h. Spots were developed according to the manufacturer’s instructions. Spots were scanned and quantified using a CTL-ImmunoSpot S5 reader. The number of spot-forming cells (SFC) was calculated by subtracting the mean number of PBS–stimulated wells. A mix of phorbol 12-myristate 13-acetate (PM, 20 ng/mL) and ionomycin (Sigma-Aldrich, 1 μg/mL) served as the positive control, with PBS used as the negative control.

### 2.13. Intracellular Cytokine Staining (ICS) and Flow Cytometry

gE-specific CD4^+^ and CD8^+^ T cells expressing IFN-γ or IL-2 were detected using ICS and flow cytometry, as previously described [35].Single spleen cells (2 × 10^6^ cells) were isolated from immunized mice and restimulated in vitro over 6 h using a pool of the 15-mer VZV–gE peptides with an 11 amino acid overlap (2 μg/mL), as described earlier. For intracellular cytokine staining, cells were incubated for 6 h with protein transport inhibitors (BD GolgiPlug; BD Biosciences). Cells were then washed with FCS (PBS containing 1% fetal calf serum) and blocked with Mouse BD Fc block at 4 °C for 30 min. After incubation, cells were stained for 30 min with a mix of PE/Cy7-conjugated anti-mouse CD8 (1:100 dilution; Biolegend, San Diego, CA, USA), FITC-conjugated anti-mouse CD4 (1:100 final dilution; Biolegend) and Live/Dead Fixable Aqua (Invitrogen; Carlsbad, CA) in a total volume of 50 μL. Cells were then fixed and permeabilized with Fixation/Permeabilization Solution Kit (BD Biosciences) and further stained with APC-conjugated anti-mouse IFN-γ and PE-conjugated anti-mouse IL-2 (1:100 final dilution; Biolegend). Cells were then washed twice with 1×Perm Wash solution, resuspended in FCS, and then analyzed using a BD LSRFortessa X-20 Flow Cytometer (Becton Dickinson). Live cells were gated (FSC/SSC) and acquisition was performed on ∼50,000 events. Data were analyzed using FlowJo software 10.4.2 (FlowJo, LLC, Ashland, OR, USA). Data are represented as background subtracted from the mean responses of gE-specific CD4^+^ T cells, expressed as percentages of the total frequencies of CD4^+^ or CD8^+^ T cells expressing IFN-γ or IL-2.

### 2.14. Statistical Analysis

Statistical analysis and graphing were done with Graphpad Prism. The statistical details are described in the figure legends. Data were presented as mean ± SD. A value of *p* < 0.05 was considered to be statistically significant and represented as asterisk (*). Value of *p* < 0.01 was considered to be more statistically significant and represented as double asterisks (**). Value of *p* < 0.001 was considered to be the most statistically significant and represented as triple asterisks (***). Value of *p* < 0.0001 was considered to be the extremely statistically significant and represented as quadruple asterisks (****). For comparison between each group with the mean of every other group within a dataset containing more than two groups, one-way ANOVA with Tukey’s test, or Kruskal–Wallis ANOVA with Dunn’s post hoc test were used as appropriate.

## 3. Results

### 3.1. Generation of a Recombinant Baculovirus Displaying the gE Protein

To generate a recombinant baculovirus displaying gE, the engineered baculovirus was constructed following the protocol provided by a well-established commercial kit. In this construction strategy, the baculovirus genome was linearized with the partial knock-out of two functional genes encoded by ORF603 and ORF1629 but having two recombination sites at its termini, then a donor plasmid carrying intact ORF603 and ORF1629 genes as well as interest gene (Figure 1A) was allowed to recombine with the linearized genome and rescue the genome with the ability to produce infectious baculoviruses. The pTriex-gE construct and blank pTriex vector were then co-transfected with baculovirus DNA into Sf9 cells for the generation and amplification of recombinant baculoviruses (Figure 1B), hereafter referred to as Bac–gE and Bac–WT, respectively. Supernatants were collected and purified to obtain high titers of the recombinant baculovirus stocks.

gE expression in Sf9 cells was confirmed via Western blotting (WB) and immunostaining using a gE-specific antibody (1B11 MAb). A positive signal for gE was identified at the expected size of ~70 KDa in the immunoblot. No detectable signal was observed in the lysates of uninfected cells or among cells infected with Bac–WT (Figure 2A).

Next, gE-positive immunofluorescence staining and flow cytometry were performed 24 h after infection to confirm gE membrane localization. As anticipated, positive fluorescence for gE was found on the membranes of Bac–gE-infected cells but not on the control cells (Figure 2C). Consistently, in the flow cytometry analysis, the ratio of gE-positive cells among the Bac–gE-infected cells was ~34.3%, which is significantly higher than 0.057% detected for the Bac–WT (control) cells.

Following this, to verify correct display of gE on the recombinant baculovirus, virus particles were concentrated by ultracentrifugation and imaged by transmission electron microscopy (TEM) (Figure 2D,E). Large rod-shaped particles were clearly observed, with an average length of 300 nm and diameter of ~25 nm. After two rounds of centrifugation, the envelope of the baculovirus remained intact (Figure 2E). There was no difference in the morphology between the Bac–gE and Bac–WT recombinant viruses (Figure 2E). Furthermore, using immune colloidal gold staining, we found the gold particles to be mainly distributed on one side of the membrane in the Bac–gE baculovirus virions (Figure 2D), with no positive signals detected in Bac–WT baculovirus virions. Collectively, these data confirm the successful expression of gE in Sf9 cells and proper display of the protein on the baculovirus particle surface.

We introduced the CMV promoter into the baculovirus vector to drive the endogenous expression of gE after transduction, which thus allowed the vector to function like a DNA vaccine. gE expression was verified by transducing the purified recombinant baculovirus (Bac–gE) into HepG2 cells at different multiplicity of infection (MOI) and then testing for expression using immunofluorescence. As anticipated, 48 h after infection, we detected positive signals indicative of appropriate gE protein expression in Bac–gE transduced cells (Figure 3). This demonstrated that Bac–gE can be successfully expressed in mammalian cells.

### 3.2. Bac–gE Elicits a Robust Humoral Immune Response in Mice

To evaluate the immunogenicity of the Bac–gE recombinant baculovirus as compared with the two other strategies, Balb/c mice (n = 5, per group) were split into six groups and administered with an intramuscular injection of Bac–gE recombinant baculovirus (with or without aluminum adjuvant [+/−Al]); recombinant gE protein with aluminum adjuvant (+Al); the v-Oka strain; Bac–WT (control 1); or saline (control 2). Aside from the control groups, each group received the same level of gE protein (0.3 μg) (quantified by a double-sandwich ELISA, R^2^ = 0.9984, Appendix A) (Table 1). Three weeks after primary immunization, the mice received a booster dose of the same amount. Sera were collected weekly to construct a time-course analysis (Figure 4A). IgG seroconversion occurred at 2 weeks after the prime (Figure 4B). We found the IgG titers to be significantly increased (~5-log) within one week after the boost immunization (4th week) (Figure 4B,C) but found no significant difference in the IgG titers between the Bac–gE and Bac–gE+Al group at the 4th week; this is not unexpected, given that the baculovirus can act as its own adjuvant. The IgG titers of Bac–gE and Bac–gE + Al groups were slightly higher than those measured for the gE+Al and v-Oka groups. As anticipated, there were no detectable VZV–specific IgG titers in mice immunized with Bac–WT or saline (Figure 4C).

We continued to assess the level of vaccine-induced immunity at four weeks after the final immunization (7th week) and found that the antibody titers remained above ~4-log without demonstrating an obvious decline. Mice immunized with Bac–gE maintained a high gE-specific IgG titer over the full 7 weeks as compared with the other groups (Figure 4B).

We next performed a plaque reduction neutralization assay to further examine the level of neutralizing antibodies induced by the four different vaccination strategies, as described elsewhere. We found that sera from Bac–gE–immunized mice could effectively protect ARPE-19 cells from infection with the v-Oka virus. As shown in Figure 4D, the plaques were obviously reduced after the sera were diluted, with mean values of ~1500-fold and ~1000-fold for Bac–gE+Al and Bac–gE, respectively. The neutralizing titers of the sera from these two groups were significantly higher than those from gE+Al and v-Oka treatments. Notably, mice immunized with gE and the v-Oka vaccine developed lower titers of neutralizing antibodies despite having a similar level of gE-specific IgG antibodies. These data clearly demonstrate that displaying the gE protein on the baculovirus surface enhances immunogenicity and induces a robust humoral immune response in mice.

### 3.3. Bac–gE Induces a Strong Cellular Immune Response in Mice

Given the important role T cells play in controlling shingles reactivation, we next examined the CMI response triggered by Bac–gE as compared with the other vaccine strategies. For this, we first assessed how the profile of the T cell compartment changes among the different groups using cellular cytokine staining. Single spleen cells were stimulated or not with the 15-mer peptides of VZV–gE with an 11-amino acid overlap covering the entire sequence of the gE protein. Flow cytometry was used to measure the frequencies of CD4^+^ and CD8^+^ T cells expressing gE-specific cytokines (IFN-γ, IL-2) (Figure 5A–D). Intriguingly, we found that CD4^+^ T cell responses were enhanced with Bac–gE without the adjuvant as compared with Bac–gE+Al (Figure 5E,F); these results suggest that the presence of aluminum might decrease the IFN-γ-associated CD4^+^ T cell response yet, at the same time, induce a strong B cell response. As compared with v-Oka and gE+Al, Bac–gE induced higher counts of gE-specific CD4^+^ T cells that were positive for both IL-2 and IFN-γ: specifically, 4- and 6-fold higher counts of IL-2-producing CD4^+^ T cells (Figure 5E), and 1.5- and 3-fold higher counts of IFN-γ-producing CD4^+^ T cells (Figure 5F). Intriguingly, Bac–gE also induced activation of cytotoxic T lymphocytes in the spleen, as evidenced by the significantly higher numbers of IL2-producing CD8^+^ T cells (Figure 5G); there was no significant change in IFN-γ-producing CD8^+^ T cell levels (Figure 5H).

We next used an ELISPOT assay and 15-mer peptides of VZV–gE with mixed peptides spanning the gE protein to compare the three vaccine modalities (we only compared Bac–gE without alum, recombinant gE protein, and live-attenuated viral vaccine, based on previous results). We found an enhanced secretion of IFN-γ lymphocytes from the spleens of immunized mice that had also received a booster dose after two weeks (Figure 6A,C). Mice immunized with Bac–gE baculovirus had higher IFN-γ responses (100–650 spots per 10^5^ cells) than did mice immunized with gE+Al (100–200 spots per 10^5^ cells) and slightly lower responses than those produced with the v-Oka vaccine (400–700 spots per 10^5^ cells) (Figure 6C). There was no difference in the responses of mice when comparing stimulated (mixed pool of gE peptides) versus unstimulated groups for IL-2 lymphocytes (Figure 6B), but overall, Bac–gE group were observed more IL-2 positive lymphocytes. This phenomenon may be related to the vector of baculovirus. Collectively, these data highlight Bac–gE as a strong inducer of a robust CMI response in mice.

## 4. Discussion

Varicella–zoster virus is a widespread human pathogen, with high rates of infection world-wide. VZV–specific T-cell immunity—particularly, the CD4^+^ T cell response—is a critical determinant of latent reactivation and disease potential and thus remains an important consideration for effective viral control [36]. The increase in VZV reactivation observed with age and the resultant rash and pain linked with the development of HZ cause a substantial societal burden [37]. Consequently, the design of an effective vaccine that can control VZV/HZ is an important goal.

To date, recombinant protein vaccines have failed to generate a sufficient and effective immune response when administered alone, and thus they rely on a strong adjuvant to improve immunogenicity and induce T cell responses [38,39]. The most effective HZ vaccine, Shingrix, comprises the subunit protein gE combined with the AS01B adjuvant. This adjuvant contains two immunostimulants, QS21 and 3-O-desacyl-4′-monophosphoryl lipid A (MPL), which induce robust CD4^+^ T cell activation and higher levels of IFN-γ and/or IL-2 [40]. Alternatively, CpG oligodeoxynucleotides have been approved by the Food and Drug Administration as safe and effective adjuvants for clinical applications [41]. CpG oligodeoxynucleotides, when used as an adjuvant, can directly and efficiently stimulate natural killer cells, dendritic cells (DC) and macrophages through Toll-like receptor (TLR) stimulation [42,43]. In contrast, baculoviruses themselves are also regarded as effective adjuvants due to the abundance of CpG elements in their genome [44] and can activate innate immunity and induce adaptive immunity; indeed, in animal models, baculovirus surface-displayed vaccines induce powerful protective immunity [45,46]. In this study, we constructed a gE-displaying baculovirus and showed that, in mice, this recombinant baculovirus evoked better humoral and cellular responses with stronger nAbs titers than the gE protein formulated with aluminum or the v-Oka vaccine. Thus, the Bac–gE vaccine can enhance immunogenicity associated with a B cell response. Further, we measured higher levels of IFN-γ and IL-2 by CD4^+^ T cells, indicating that purified Bac–gE can act as an immunogen without the need for an adjuvant.

Baculoviruses can induce dendritic cell (DC) maturation, and activate antigen-presenting cells (APCs), which in turn, release high numbers of cytokines through MyD88/TLR9-dependent signaling [47]. Previous studies have shown a transient elevation in the IFN and TNF pathways, which may shape the adaptive immune response [48].These innate immune factors may be particularly advantageous, as the innate immune response can affect the speed of the adaptive immune response. Indeed, monocytes and macrophages can acquire memory that is specific for particular major histocompatibility complex I antigens using paired A-type immunoglobulin-like receptors (PIR-As) [49]. These adaptive changes will be important considerations in future studies.

In various types of infectious disease, CD8^+^ T cells function to eliminate virally infected cells and their potent cytotoxic T lymphocyte (CTL) responses. MHC class I antigens present endogenous or intracellular peptides that are recognizable by CD8^+^ T cells. Accumulating evidence suggests that it is difficult to mount robust CD8^+^ T cell responses using subunit protein vaccines [50,51,52]. Yet, in our study, we detected robust CD8^+^ T cell populations, providing potential evidence for MHC-I antigenic presentation of the gE protein under the control of CMV promoter. Shingrix has been shown to induce a sufficient CD4^+^ T cell response but a lower CD8^+^ T cell response. However, in our strategy, Bac–gE acquired the advantages of both a virus-like particle vaccine and a DNA-based vaccine, eliciting robust CD4^+^ and CD8^+^ T cell responses. Overall, the baculovirus is an effective alternative vector vaccine for diseases wherein a T cell response is critical for virus clearance, such as VZV reactivation.

Overall, we show that, as a viral vaccine vector, baculoviruses offer several unique properties for vaccine development: (1) there is no preexisting immunity against baculoviruses in humans; (2) baculoviruses can accommodate large exogenous DNA fragments; (3) baculoviruses generate high titers of virus stock at a low-cost and with a labor-saving strategy; (4) baculoviruses induce robust innate immunity and adaptive immunity, without the need for any additional adjuvant [53]. These advantages render Bac–gE a promising vaccine candidate for HZ and other related diseases, particularly among immunocompromised individuals. Nevertheless, this study has two apparent limitations that need to be clarified in future. First, due to the high host specificity of VZV, v-Oka vaccine suffered from limited immunogenicity in a mouse model. Second, the immunogenicity of gE-displaying baculoviruses should be directly compared with the commercial Shingrix that has been approved with excellent HZ protection efficacy, and the safety profile should be comprehensively evaluated in systematic animal experiments. In the meantime, the various processes associated with the design and production of a baculovirus-based vector vaccine, such as clarification, concentration, purification and formulation, must be addressed and thoroughly investigated to fulfill the requirements of clinical studies.

## Figures and Tables

**Figure 1 viruses-14-01785-f001:**
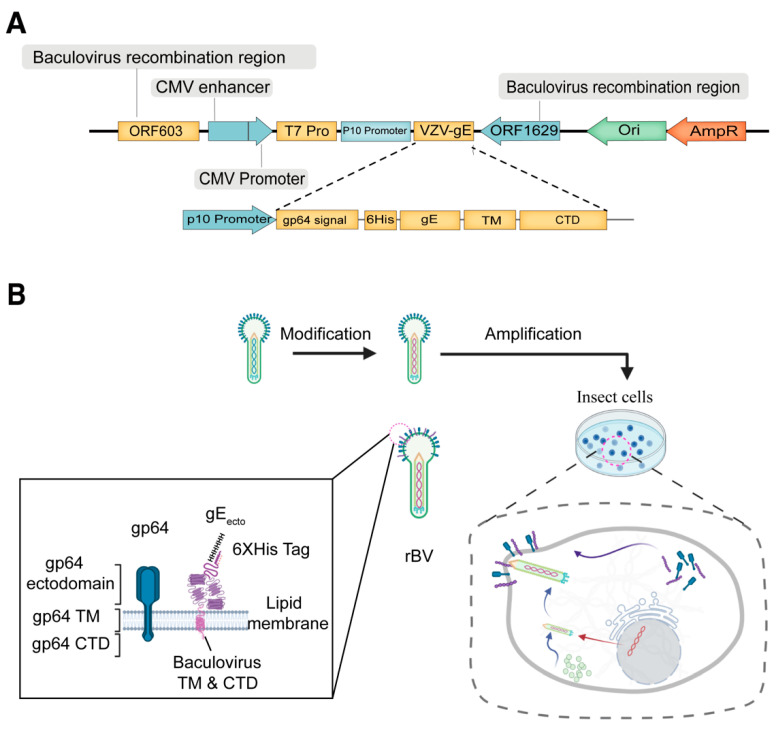
Construction of recombinant baculovirus displaying the gE protein. (**A**) A schematic illustration of the Varicella zoster virus (VZV) gE protein recombinant expression cassette. The coding sequences of the VZV–gE gene were cloned into the baculovirus vector, pTriEx-5, which contains the gene for the gp64 signal peptide (SP), a 6×His tag in the N-terminus, a transmembrane domain (TM) of gp64, and the cytoplasmic domain (CTD) of gp64 at the C-terminus of the cassette. (**B**) Flow chart of the budding process and amplification of the recombinant baculovirus (rBV) Bac–gE in Sf9 insect cells. The gE protein is displayed on the lipid membranes of cells. High titers of rBV are produced following several rounds of infection in Sf9 cells.

**Figure 2 viruses-14-01785-f002:**
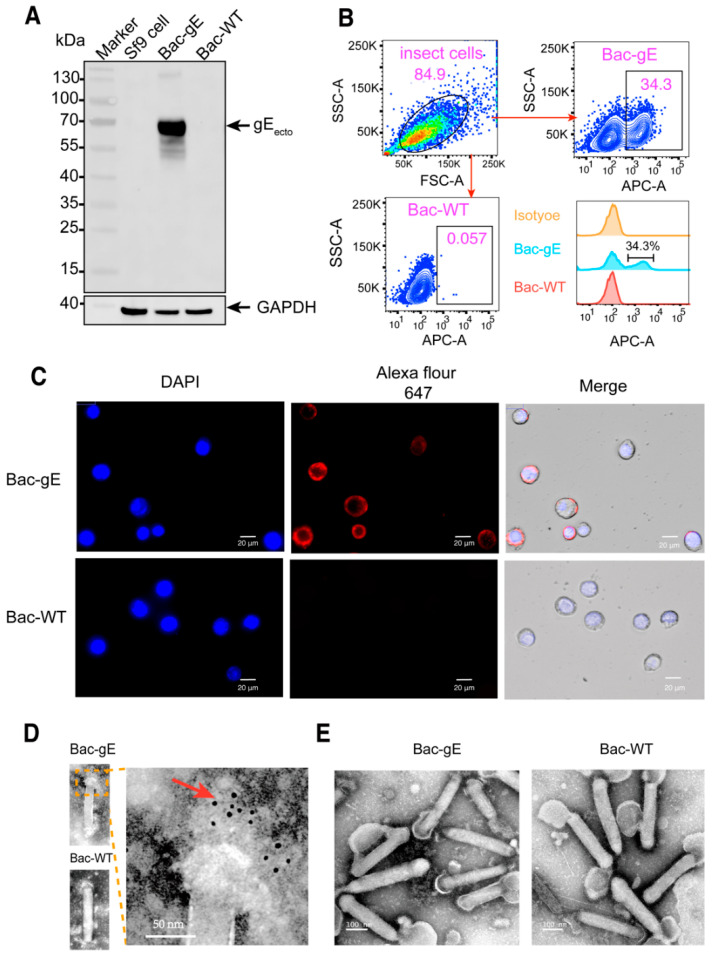
Characterization and generation of the displayed gE proteins. (**A**) Western blot analysis of gE protein expression by Sf9 cells infected with the recombinant baculovirus (Bac–gE, or Bac–WT as a control). Protein was detected using the anti-His primary antibody and repeated twice. (**B**) Frequency of gE expression in cells by flow cytometry analysis. Positive signals indicate successful display of the gE protein on the cell membrane. Three experiments were conducted, and the data presented represent the results from one independent infection experiment. (**C**) Immunofluorescence assay for the expression of gE proteins in Sf9 cells. Sf9 cells were infected with Bac–gE and Bac–WT and then fixed, stained, and analyzed using fluorescence microscopy (bar = 20 μm). Red fluorescence suggests the expression of gE. Blue fluorescence (DAPI) staining is used to indicate the location of the nucleus. (**D**) Electron micrographs of immunogold-labeled rBV Bac–gE. Positive colloid gold signals indicated by the red arrow showed that VZV gE proteins are displayed on the surface of the recombinant baculovirus. (bar = 50 nm). (**E**) Electron microscopic images of negatively stained wild-type and gE baculovirus particles (bar = 100 nm). Transmission electron microscopy experiments were conducted twice. The data presented are from one representative experiment.

**Figure 3 viruses-14-01785-f003:**
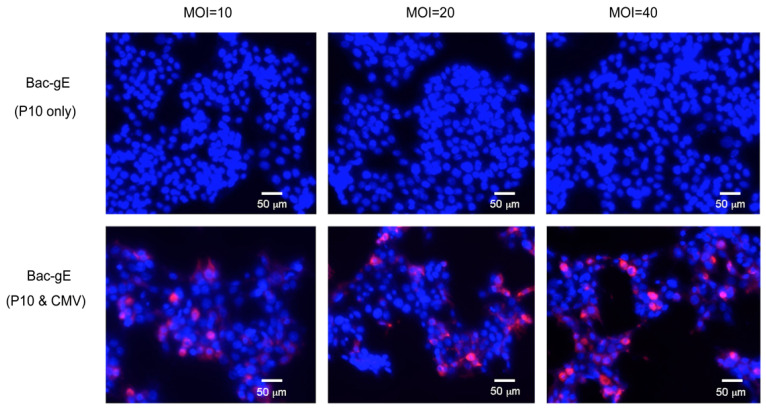
The expression of gE in mammalian cells was tested by transducing HepG2 cells with rBV Bac–gE (carrying P10 only or P10 and CMV promotors) at different multiplicity of infection (MOI). Red fluorescence suggests the expression of gE in HepG2 cells; blue fluorescence (DAPI) indicates the location of the nucleus (bar = 50 μm). Four technical replicates were performed during one experiment.

**Figure 4 viruses-14-01785-f004:**
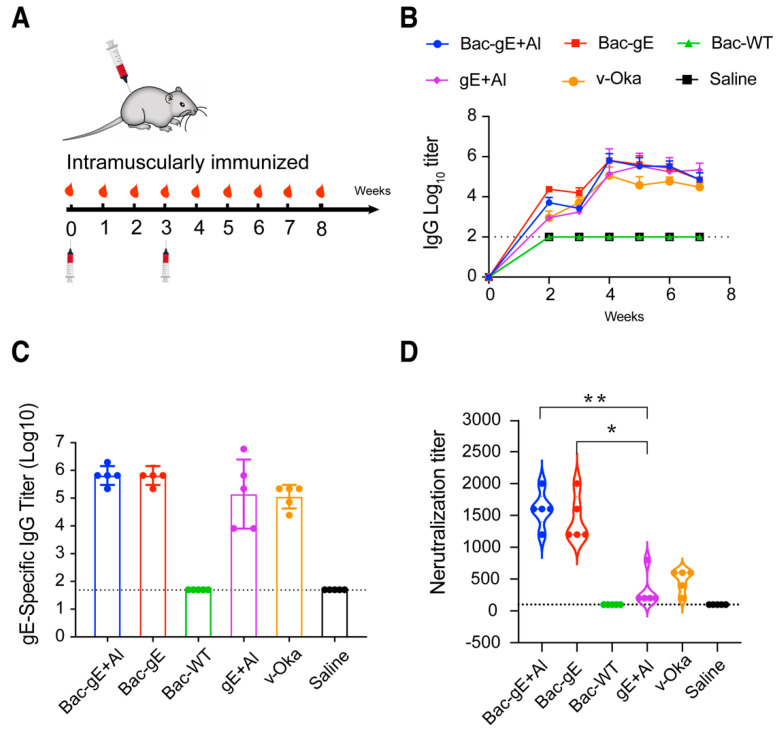
Humoral immune response elicited by Bac–gE in mice. (**A**) Immunization procedure of different VZV vaccine modalities in Balb/c mice. (**B**) gE-specific IgG antibody titers taken each week were determined by indirect ELISA. (**C**) gE-specific IgG antibody titers at week 4 after two rounds of immunization (date was shown as mean ± SD, n = 5). (**D**) Neutralizing antibody titers determined by plaque reduction neutralization assay after boost immunization and plotted as the Violin plot (show all point). One-way ANOVA with Kruskal–Wallis test was used for inter-group statistical comparison. * *p* < 0.1, ** *p* < 0.01. Two technical replicates were performed during one experiment.

**Figure 5 viruses-14-01785-f005:**
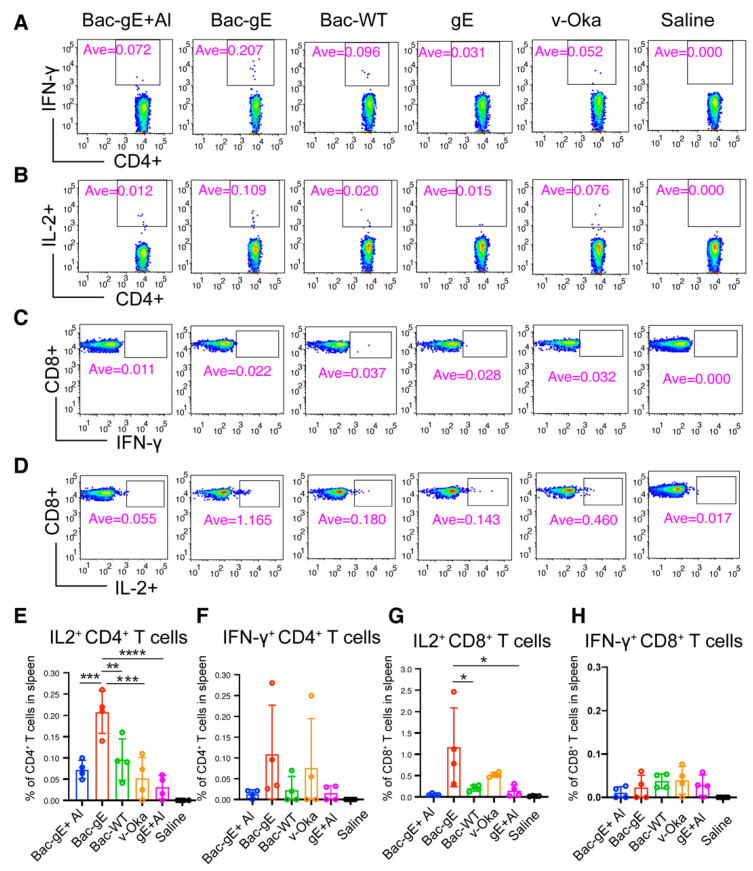
Intracellular cytokine staining (ICS) to quantify key cytokine production following VZV gE stimulation. (**A**–**D**) Representative flow cytometry images following treatment with one of the four vaccine modalities. Pseudo-color dot-plots showing T cell subsets [IFN-γ-producing CD4+ T cells] (**A**), [IL-2-producing CD4^+^ T cells] (**B**), [IFN-γ-producing CD8^+^ T cells] (**C**) and [IL-2-producing CD8^+^ T cells] (**D**) were identified by intracellular cytokine staining (**E**–**H**). Positive cytokine production by CD4^+^ and CD8^+^ cells in response to VZV gE-peptide stimulation. Histograms represent the frequency of IL-2-positive CD4^+^ cells (**E**), IFN-γ-positive CD4^+^ cells (**F**), IL-2-positive CD8^+^ cells (**G**) and IFN-γ-positive CD8^+^ cells (**H**) following VZV gE-peptides stimulation. Values are shown as mean ± SD (n = 4). One-way ANOVA with Tukey’s multiple comparisons test was used for inter-group statistical comparisons. * *p* < 0.1, ** *p* < 0.01, *** *p* < 0.001, **** *p* < 0.0001. Flow cytometry data were obtained from a single experiment.

**Figure 6 viruses-14-01785-f006:**
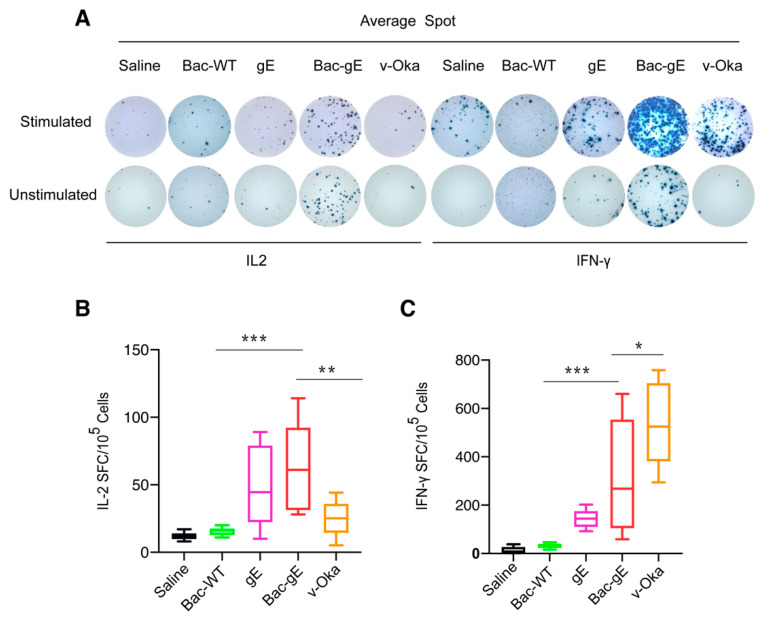
Elispot assay to evaluate splenocyte secretion of IFN following VZV gE stimulation. (**A**) Elispot representative images obtained with or without stimulation with one of the four vaccine modalities. (**B**,**C**) Quantitative measurements of the number of IL-secreting (**B**) and IFN-secreting (**C**) cells, respectively. Results are reported as number of spot-forming cells (SFCs). Data are plotted as box and whiskers. In box and whisker plots, data are presented as min to max whiskers and the median indicated by a horizontal line. One-way ANOVA with Tukey’s multiple comparisons test was used for inter-group statistical comparisons. * *p* < 0.1, ** *p* < 0.01, *** *p* < 0.001. One independent experiment was conducted for Elispot analyses.

**Table 1 viruses-14-01785-t001:** Mouse Immunization Study Design.

Vaccine	Dose (PFU)	Dose (ug)	Bleeding (Week)	Timepoint Spleen Removal (Week)
Bac–gE +Al	1 × 10^8^	0.3	1–7	5
Bac–gE	1 × 10^8^	0.3	1–7	5
Bac–WT	1 × 10^8^	-	1–7	5
gE+Al	-	0.3	1–7	5
Voka	5 × 10^6^	0.3	1–7	5
PBS	-	-	1–7	5

## Data Availability

The published article includes all datasets generated or analyzed during this study.

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
