# Peer review of "Baculovirus Display of Varicella–Zoster Virus Glycoprotein E Induces Robust Humoral and Cellular Immune Responses in Mice"

_viruses, 2022, doi:10.3390/v14081785_

Round 1
Reviewer 1 Report
Wenhui Xue et al, have addressed the comments from the reviewers. The manuscript looks much more improved. The inclusion of additional experiments and updates to the existing experiments have strengthened the overall scientific value of the manuscript.
Author Response
Point 1: Wenhui Xue et al, have addressed the comments from the reviewers. The manuscript looks much more improved. The inclusion of additional experiments and updates to the existing experiments have strengthened the overall scientific value of the manuscript.
Response 1: We appreciate your effort in reviewing our manuscript. With your help and wise suggestion, our manuscript was significantly improved.
Reviewer 2 Report
The authors have satisfactorily improved the manuscript based on the comments provided by the reviewers with the exception of one major concern:
It remains unclear (perhaps unconvincing) to readers whether each experiment was repeated (or not). The authors' disclosure indicates that all relevant data are included in the manuscript, yet, only one experiment is ever shown. In the pursuit of scientific rigor, even if data reflect one representative experiment, the statistics section or each respective figure legend must indicate how many total experiments were performed for each assay. Addition of this element is critical for publication.
Author Response
Point 1: The authors have satisfactorily improved the manuscript based on the comments provided by the reviewers with the exception of one major concern:
It remains unclear (perhaps unconvincing) to readers whether each experiment was repeated (or not). The authors' disclosure indicates that all relevant data are included in the manuscript, yet, only one experiment is ever shown. In the pursuit of scientific rigor, even if data reflect one representative experiment, the statistics section or each respective figure legend must indicate how many total experiments were performed for each assay. Addition of this element is critical for publication.
Response 1: Sorry for missing the information. In this study, all experiments, except for the immunization assay, were repeated at least two times and showed consistent results. Therefore, data are presented from one representative experiment. The relavant information has been added in all Figure legends in the revised manuscript.
This manuscript is a resubmission of an earlier submission. The following is a list of the peer review reports and author responses from that submission.
Round 1
Reviewer 1 Report
Wenhui Xue et al, have described the engineering of Baculovirus to express VZV glycoprotein E. This is an interesting premise, and the efficacy of engineered viruses in relation to VZV humoral and cellular immune response has been picturesque, using simple but effective experiments. The results are intriguing and potentially quite exciting. However, there are a few important points that need to be addressed.
Major comments:
1) A high-resolution image of the gE expression in Sf9 cells after infection with Bac-gE, to show the gE membrane localization is needed, as the image (Fig 2C) provided didn’t clarify specifically localization, as the merged images appear to be restricted to the nucleolar.
2) The authors introduced the CMV promoter into the baculovirus vector, for the endogenous expression of gE, favoring vector mediated DNA vaccine approach. But the transduction into HepG2 does not show, a significant population of cells being positive for gE protein expression (Fig 2F). The experiment didn’t clarify the MOI of Bac-gE used for the transduction of the HepG2 cells. Maybe a dose-dependent transduction approach will shed light on the raised question.
Minor comments:
1) [ Lines 67-69] Relevant citations should be added.
2) [ Lines 95-96] Relevant citations should be added.
3) [ Line 131] Replace ‘Alter’ with After.
4) In the Materials and methods section 2.4 Western blotting. Provide information on the housekeeping gene/ loading control.
5) Correction should be made to the labeling in the panel (A) referring to the loading control “NAPDH”.
6) Provide information about the RED arrow used in panel D of (Fig 2), in the legend.
7) Figures 4 & 5 are not specifically described in the legend.
Reviewer 2 Report
This is the first nonclinical study to evaluate immunogenicity of baculovirus based VZV glycoprotein E vaccine in mice. Although VZV infects only humans, however, immunogenicity of other HZ vaccine was evaluated using mice.
1. I think it would be good for the authors of this study to describe why a new herpes zoster vaccine platform is needed, in the introduction or discussion section. Because recently approved HZ vaccine, Shingrix showed superior efficacy than live attenuated zoster vaccine. 2. The authors compared immunogenicity of various vaccines in mice. Humoral immunity of Bac-gE was superior gE+Al, and ICS showed that CMI by Bac-Ge was superior than gE+Al. Shingrix used adjuvant as AS01B. Is there any data comparing Bac-gE and Ge-AS01B?3. There is no mention of limitations of this study in the current manuscript. Are there really no limitations or weak points of this study?
Reviewer 3 Report
Major concern:
1. Single plasmids containing single promoter should be added as controls to prove the DNA/protein vaccine effect.
2. Error bars too high in flow cytometry analysis; inconsistency between results from flow cytometry and ELISPOT analysis made the conclusions untrustable.
Minor concerns:
1. Introduction was poorly written. For example, VZV is a virus instead of a disease, and not the 90% infected people must have chickenpox (lines 39-40). Deddritic cells were not expressed (lines 92-93). Humans of course have immune responses to baculoviruses (line 96).
2. Discussion was poorly written without proper citation of previously published corresponding research.
Reviewer 4 Report
In this manuscript submitted to Viruses, Xue and colleagues present the characterization of an experimental vaccine for varicella-zoster virus using an innovative baculovirus expression vector approach. Overall, the manuscript is presented in a clear and logical fashion; the studies are generally well designed and provide verification of the vector construct fidelity and immunogenicity. That said, several issues must be addressed to improve clarity of the manuscript.
General comments:
· The introduction is too broad and long. Consider citation of a relevant review on select topics to make this section more concise. Perhaps focusing on one or two points that address the clinical need for yet another VZV vaccine would be most useful. Also, ensure relevance and authority or timeliness of references.
· One important issue to address is that Shingrix is only utilized as a therapeutic vaccine in VZV-positive human adults. The experimental data within the manuscript only demonstrate the immune responses in naive mice. How should this impact interpretation of the data with respect to any clinical implications?
· The methods should include a statistics section. The statistical analyses reported (Mann-Whitney U) are not appropriate for comparisons of three or more groups. Please thoroughly review and reanalyze data for all figures using ANOVA with an appropriate post-hoc test. Furthermore, it is imperative that the n values and number of experiments are reported. Do the graphs reflect one representative experiment or aggregate data from multiple experiments?
· The Shingrix vaccine uses the AS01 “robust adjuvant” system that includes monophosphoryl lipid A (CD4 T cell stimulant). Why were the experimental subunit vaccines administered with alum only for experiments herein? This doesn’t seem like a straightforward comparison, given the increased antigenicity of AS01.
· Methods section 2.9: Please justify why only female mice were used. Inclusion of immunogenicity data in males would be insightful given the importance of sex as a biological variable.
Specific comments:
· Ideally, the title should indicate that the immunogenicity studies were conducted “in mice.” At a minimum, the abstract must include the species in which the vaccine was tested.
· Abstract line 17: Replace “causing infection” with “HZ.”
· Lines 46-47: Replace “In global” with “Worldwide” and capitalize
“China.”
· Lines 67-68: “Despite the efficacy, Shingrix…” Please add a citation to validate this statement.
· Line 86: Reference #26 is is over a decade old and out of date. Clinicaltrials.gov lists influenza and COVID-19 as infectious diseases for which one or more clinical trials are utilizing baculovirus vaccines.
· Line 90: Reference 27 does not validate the CpG content of baculoviruses, and reference 28 appears irrelevant. Consider adding the following reference regarding the CpG content of baculovirus: Abe T, Hemmi H, Miyamoto H, Moriishi K, Tamura S, et al. (2005) Involvement of the Toll-like receptor 9 signaling pathway in the induction of innate immunity by baculovirus. J Virol 79: 2847–2858.
· Methods section 2.6: Would it be important to permeabilize the Sf9 cells prior to labeling for accurate verification of protein expression by immunofluorescence/flow cytometry?
· Methods section 3.2 (lines 335-337). More detail is required to verify how gE production was assessed and normalized for the purpose of vaccination. What cells were used to determine the amount of gE produced using Bac-gE and v-Oka? Was this in vitro? If so, describe explicitly in the methods. As it stands, the gE dose (μg) outlined in Table 1 is difficult to interpret.
· Line 352: The sentence beginning with “The IgG titers” must indicate “gE-specific” here. The v-Oka has greater antigenic breadth. Accordingly, this is not necessarily a fair comparison. The line also indicates a difference between groups (“slightly higher”) that is not corroborated by a statistical comparison/difference in Figure 3.
· Panels A and B do not add valuable information to Figure 3. Consider adding the R-squared value as a comment in the text only.
Reviewer 5 Report
This manuscript describes the generation of a recombinant baculovirus that expresses VZV gE from a multifunctional promoter. The purpose of making this rBV Bac-gE was to evaluate it as a novel vaccine platform. The B and T cell responses to rBV Bac-gE were compared to adjuvanted gE protein and the live-attenuated v-Oka vaccine strain in mice. The key results were demonstrating VZV gE expression on the head end of baculovirus, in infected insect cells, and in transduced mammalian cells. The rBV Bac-gE also generated high levels of gE-specific IgG that neutralized virus better than gE protein or v-Oka. The rBV Bac-gE was better than v-Oka at eliciting CD4+ and CD8+ T cells that expressed IL-2, whereas these vaccines were equivalent at eliciting T cells that expressed IFN-g. This is an interesting study and could lead to a new type of vaccine for herpes zoster. There are several major issues that should be addressed, including the purity and sequence of the rBV Bac-gE, the localization of Bac-gE in mammalian cells, why Shingrix was not included, and the source of the recombinant gE protein.
Line 19, change the word “albeit” to “however”.
Line 21, change the word “engineering” to “engineered”.
Line 22, change the phrase “immune colloidal gold experiments” to “immune electron microscopy”.
Line 46, change the words “In global” to “Globally”.
Line 48, capitalize China.
Line 48, remove the hyphen in “anti-virals”.
Line 49, change the spelling of “valacyclovir” to “valaciclovir”.
Line 64-65, change the phrase “capable of eliciting” to “that elicits”.
Line 69, remove the “s” from the word “receptors”.
Line 78, insert the word “multicapsid” before “nucleopolyhedrovirus”.
Line 86, italicize “Plasmodium berghei”.
Line 96, clarify the statement that there have been no reported immune responses against baculoviruses in humans. Does this mean that humans do not generate immune responses against baculoviruses in the environment? In engineered vaccines? It is important to explain whether immune responses against a recombinant baculovirus vaccine are possible, and if they pose a problem for using this platform for multiple antigens or vaccines. For instance, will a person generate immune responses against baculovirus antigens in a rBV vaccine? Will this interfere with subsequent rBV vaccines for different antigens? Boosters? Or will the immune responses to rBV vaccines improve the efficacy against the recombinant antigen?
Line 110, italicize “Spodoptera frugiperda”.
Line 131, correct the typo in “Alter”. It should be “After”.
Line 169, change format of the exponent to superscript in “2x106”.
Line 179, change the words “seeded into” to “coated onto”.
Line 188, change “into” to “onto”.
Line 193, change the chemical name of “H2SO4” to include subscript numbers (H2SO4).
Line 211, how was the amount of gE in the v-Oka vaccine measured? This vaccine is a live-attenuated virus preparation, so the gE is part of the virions and infected cells. Were the proteins released from the cells and virions before performing the sandwich ELISA? If not, then the amount of gE was likely very much higher than 0.3 micrograms.
Line 245, change format of the exponent to superscript in “2x106”.
Fig. 1, the text and the figure caption describe a 6xHis tag at the N-terminus of gE, but the diagram does not show it. Include the His tag in the gene diagram in (A) and the drawing enlargement in (B).
Line 286, gE protein expression was verified by several methods, but the recombinant baculovirus was not analyzed for the sequence of the insert. Justify why the virus was not plaque purified, then sequenced to confirm the homogeneity of the recombinant strain and the correct arrangement of the gE cassette. This is especially important for vaccines, which must be genotyped. It is possible that the high titer rBV stock is a mixture of WT and gE-expressing genomes. This seems likely because the flow cytometry assay measured 34.3% of infected cells expressing gE. This is a major drawback of this study.
Fig. 2, the images in 2F are not acceptable. Higher magnification images are necessary to observe gE expression following rBV Bac-gE transduction into HepG2 cells. Since this chimeric gE lacks its own cytoplasmic tail, it may not localize to the same sites as WT gE. It would be helpful to compare Bac-gE expression and localization to v-Oka gE in these cells. A magnification of 40X is the minimum needed to evaluate this question, and a measurement bar should be placed on the figure.
Line 334, what was the source of recombinant gE protein? This requires a citation, a source, or a detailed section in the Methods. There is no mention of this reagent in the Methods.
Line 337, justify why this project did not include Shingrix as one of the vaccine modalities. It is the most similar to rBV Bac-gE. In addition, clarify whether the v-Oka preparation used was the varicella vaccine or a zoster vaccine. Varivax (Merck) is a lower titer than Zostavax (Merck). Is that true of the Wantai product? The live v-Oka vaccine is not likely to be strongly immunogenic in mice because it does not replicate in mouse cells or become latent in mouse neurons. This should be mentioned in the Discussion.
Fig. 3, panels A and B are not necessary and should be removed. The R2 value can be written into the text of the Methods. What do the error bars indicate? Standard deviation or standard error?
Fig. 4, what do the error bars indicate? Standard deviation or standard error?
Line 421, the sentence does not make sense and there is a typo: “but participants overall, Bac-gE group were abserved more IL-2 positive lymphocytes”.
Consider changing the color scheme in all the figures. It would help the reader if the vaccine and control groups were given the same color in all figures. For example, saline could be black in all figures and v-Oka could be gold in all figures, etc.